

# HDO and $H_2O$ total column retrievals from TROPOMI shortwave infrared measurements

Remco Scheepmaker[1], Joost aan de Brugh[1], Haili Hu[1], Tobias Borsdorff[1], Christian Frankenberg[2], Camille Risi[3], Otto Hasekamp[1], Ilse Aben[1], and Jochen Landgraf[1]

[1]SRON Netherlands Institute for Space Research, Utrecht, The Netherlands
[2]Jet Propulsion Laboratory (JPL), California Institute of Technology, Pasadena, CA, USA
[3]Laboratoire de Météorologie Dynamique, Insitut Pierre Simon Laplace, CNRS, Paris, France

*Correspondence to:* J. Landgraf
(J.Landgraf@sron.nl)

**Abstract.** The Tropospheric Monitoring Instrument (TROPOMI) onboard the European Space Agency Sentinel-5 Precursor mission is scheduled for launch in the last quarter of 2016. As part of its operational processing the mission will provide $CH_4$ and CO total columns using backscattered sunlight in the shortwave infrared band (2.3 μm). By adapting the CO retrieval algorithm, we

have developed a non-scattering algorithm to retrieve total column HDO and $H_2O$ from the same measurements under clear sky conditions. The isotopologue ratio HDO/$H_2O$ is a powerful diagnostic in the efforts to improve our understanding of the hydrological cycle and its role in climate change, as it provides insight in the source and transport history of water vapour, nature's strongest greenhouse gas. Due to the weak reflectivity over water surfaces we need to restrict the retrieval

to cloud-free scenes over land. We exploit a novel two-band filter technique, using strong-vs-weak water or methane absorption bands, to pre-filter scenes with medium-to-high level clouds, cirrus or aerosol and to significantly reduce processing time. Scenes with cloud top heights $\lesssim 1$ km or very low fractions of high-level clouds, or scenes with an aerosol layer above a high surface albedo are not filtered out. We use an ensemble of realistic measurement simulations for various conditions to

show the efficiency of the cloud filter and to quantify the performance of the retrieval. The single measurement precision in terms of $\delta$D is better than 15–25‰ for even the lowest surface albedo (2–4‰ for high albedos), while a small bias remains possible of up to $\sim 20$‰ due to remaining aerosol or up to $\sim 70$‰ due to remaining cloud contamination. We also present an analysis of the sensitivity towards prior assumptions, which shows that the retrieval has a small but significant sensitivity

to the a priori assumption of the atmospheric trace gas profiles. Averaging multiple measurements over time and space, however, will reduce these errors, due to the quasi-random nature of the pro-





file uncertainties. The sensitivity of the retrieval with respect to instrumental parameters within the expected instrument performance is $< 3‰$, which represents only a small contribution to the overall error budget. Spectroscopic uncertainties of the water lines, however, can have a larger and more
systematic impact on the performance of the retrieval and warrant further reassessment of the water line parameters. With TROPOMI's high radiometric sensitivity, wide swath (resulting in daily global coverage) and efficient cloud filtering, in combination with a spatial resolution of $7 \times 7$ km$^2$, we will greatly increase the amount of useful data on HDO, H$_2$O and their ratio HDO/H$_2$O. We showcase the overall performance of the retrieval algorithm and cloud filter with an accurate simulation
of TROPOMI measurements from a single overpass over parts of the USA and Mexico, based on MODIS satellite data and realistic conditions for the surface, atmosphere and chemistry (including isotopologues). This shows that TROPOMI will pave the way for new studies of the hydrological cycle, both globally and locally, on timescales of mere days and weeks instead of seasons and years, and will greatly extend the HDO/H$_2$O datasets from the SCIAMACHY and GOSAT missions.

## 35    1    Introduction

Water vapour, being the strongest natural greenhouse gas, plays a vital role in our understanding of climate change. It is part of a positive atmospheric feedback mechanism (Soden et al., 2005; Randall et al., 2007), and it plays a role in the mechanisms of cloud formation, of which the feedback mechanisms are still poorly understood (Boucher et al., 2013). A correct understanding of the many
interacting processes that control atmospheric humidity are crucial for General Circulation Models (GCMs) to come to accurate climate projections (Jouzel et al., 1987; Yoshimura et al., 2011; Risi et al., 2012a,b).

Measurements of stable water isotopologues, such as HDO, can be a unique diagnostic to improve our knowledge of the hydrological cycle (Dansgaard, 1964; Craig and Gordon, 1965). Different iso-
topologues have different equilibrium vapour pressures, which leads to a temperature dependent isotope fractionation whenever phase changes occur. The ratio HDO/H$_2$O of an air parcel is therefore dependent on the source region's location and temperature and the entire transport history of the air parcel, including all evaporation, condensation and mixing events. This makes measurements of the ratio HDO/H$_2$O a valuable benchmark for the evaluation and further development of GCMs
and explains why isotopologues have been used for decades in the fields of paleoclimatology, either using ice cores (Dansgaard et al., 1969; Jouzel et al., 1997) or speleothems (Lee et al., 2012), and hydrology in general (Mook, 2000; Aggarwal et al., 2005).

In the last decade there has been a rise in the application of water isotopologues to the atmospheric component of the hydrological cycle. This is directly related to improved remote-sensing
techniques that can accurately measure water vapour isotopologues from ground-based networks, such as the Total Carbon Column Observing Network (TCCON, Wunch et al., 2011) and the Net-



work for Detection of Atmospheric Composition Change (NDACC, formerly the Network for Detection of Stratospheric Change, Kurylo and Solomon, 1990; Schneider et al., 2016), as well as global measurements from space with instruments such as the Interferometric Monitor for Greenhouse
gases (IMG, Zakharov et al., 2004), the Thermal Emission Spectrometer (TES, Worden et al., 2007), the SCanning Imaging Absorption spectroMeter for Atmospheric CHartographY (SCIAMACHY, Frankenberg et al., 2009; Scheepmaker et al., 2015), the Infrared Atmospheric Sounding Interferometer (IASI, Herbin et al., 2009) and the Greenhouse gases Observing Satellite (GOSAT, Frankenberg et al., 2013; Boesch et al., 2013). These new techniques allow for more frequent and global
measurements of the ratio $HDO/H_2O$ in water vapour, and show the clear potential to further our understanding of the atmospheric hydrological cycle through comparisons with GCMs (Frankenberg et al., 2009; Yoshimura et al., 2011; Risi et al., 2012a,b).

Here, we present an algorithm and performance analysis for new measurements of total column HDO and $H_2O$ using the Tropospheric Monitoring Instrument (TROPOMI, Veefkind et al.,
2012), onboard the European Space Agency (ESA) Sentinel-5 Precursor (S5P) mission, scheduled for launch in Q4 2016. Like SCIAMACHY, TROPOMI will measure HDO and $H_2O$ in a 2.3 μm shortwave infrared (SWIR) band of backscattered sunlight, which provides a high sensitivity near the surface. TROPOMI, however, will have a higher spatial resolution with $7 \times 7$ km$^2$ ground pixels, better radiometric performance, a larger swath and shorter revisit time, resulting in daily global
coverage and many more measurements over cloud-free land pixels, while also demanding more efficient processing. With TROPOMI we have the opportunity to extend and improve the existing global $HDO/H_2O$ time series, and study spatial and temporal gradients with higher spatial sampling and resolution.

In Sect. 2 we describe how we adapted TROPOMI's CO algorithm to retrieve HDO, $H_2O$ and
their respective averaging kernels and how we filter for cloudy scenes. We then describe the performance of the algorithm in Sect. 3, as tested on a series of synthetic measurements with systematically varying scattering layers. A sensitivity analysis of the various input parameters is presented in Sect. 4. The performance on a realistic scenario of measurements above North America is presented in Sect. 5. Finally, in Sect. 6 we discuss our results in the context of other studies and we formulate
our conclusions.

## 2 Retrieval algorithm description

Due to the large difference in atmospheric abundance between HDO and $H_2O$, the measurement sensitivity, reflected in the averaging kernels, is very different for HDO and $H_2O$. This makes the interpretation of their ratio very challenging under conditions of light scattering by clouds. We
therefore have to pre-filter for the most cloudy conditions, which at the same time reduces processing time. This cloud filter will be described in Sect. 2.3. After cloud filtering we use a non-scattering





retrieval algorithm, adapted from the SICOR algorithm that has already been developed as part of
ESA's operational CO algorithm, which is described by Landgraf et al. (2016) in this issue. By using
this heritage of TROPOMI's CO processing, we benefit from an algorithm optimised for speed,
while also leveraging already existing expertise and software. Sections 2.1 and 2.2 describe the
specific implementation of the algorithm needed to retrieve both the $H_2O$ and HDO total column
densities.

### 2.1 Forward model and averaging kernels

Throughout this paper we consider all water isotopologues present in the SWIR range, i.e. $H_2^{16}O$,
$H_2^{18}O$ and $HD^{16}O$, as separate absorbing species. For readability, however, we will simply write
"$H_2O$" when in fact we refer to the main isotopologue $H_2^{16}O$, and "HDO" when we refer to $HD^{16}O$.
To simulate the SWIR radiance measurement, we employ a non-scattering forward model $\mathbf{F}$ that
simulates the reflected Earth radiance at its spectral sampling point $\lambda_i$ by the spectral convolution
of the simulated radiance at the top of the model atmosphere $I^{TOA}$ with the instrument spectral
response function (ISRF) $s_i$,

$$F_i = s_i * I^{TOA} \ . \tag{1}$$

Here, we assume that sunlight is scattered only at the Earth surface into the satellite line of sight
(LOS) and is attenuated by atmospheric absorption along its path. Using this approximation, the
simulated radiance at wavelength $\lambda$ is given by:

$$I^{TOA}(\lambda) = A_s(\lambda) \frac{\mu_0 F_0(\lambda)}{\pi} \exp\left(-\frac{1}{\tilde{\mu}} \tau_{tot}(\lambda)\right) \ , \tag{2}$$

where $A_s$ is the Lambertian surface albedo, $\mu_0 = \cos(\Theta_0)$ with the solar zenith angle $\Theta_0$. For low
solar zenith angles, $\mu_0$ is corrected for the sphericity of the Earth according to Kasten and Young
(1989). $F_0$ is the solar irradiance inferred from TROPOMI solar measurements (van Deelen et al.,
2007; Landgraf et al., 2016) and

$$\tilde{\mu} = \frac{\mu_0 \mu_v}{\mu_0 + \mu_v} \tag{3}$$

indicates the air mass factor with $\mu_v = \cos(\Theta_v)$ and viewing zenith angle $\Theta_v$. The total optical
thickness $\tau_{tot}$ is given by

$$\tau_{tot}(\lambda) = \sum_k \int_0^{z_{TOA}} \sigma_k(z, \lambda) \, \rho_k(z) \, dz \ , \tag{4}$$





where $z$ indicates the altitude ranging from the surface $z = 0$ to the top of the model atmosphere
$z_{\mathrm{TOA}}$. Index $k$ represents the relevant absorbers: CO and CH$_4$ including all their isotopologues,
and H$_2$O, H$_2^{18}$O and HDO. $\rho_k(z)$ is the concentration of absorber $k$ at altitude $z$ and $\sigma_k(z, \lambda)$ are
the corresponding wavelength- and altitude-dependent absorption cross sections.

The retrieval relies on a priori concentration profiles for CO and CH$_4$ from the TM5 chemistry
model (Krol et al., 2005) and specific humidity profiles from the European Centre for Medium-
Range Weather Forecast (ECMWF, Dee et al., 2011). These profiles are interpolated to the higher
resolution of the TROPOMI pixels, using a digital elevation model to account for variations in air
mass due to orography. Specific humidity is converted into concentration profiles for the absorbers
H$_2$O, H$_2^{18}$O and HDO using their natural abundance ratios.

In Fig. 1 we show a simulated transmission spectrum for the entire TROPOMI SWIR spectral
range (2305–2385 nm). The top panel shows the total transmission, while the lower four panels
show the individual transmissions of the main absorbing species (H$_2$O, HDO, CH$_4$ and CO). For
the retrieval of the HDO and H$_2$O total column densities we chose the spectral window between
2354.0–2380.5 nm (indicated with the grey band), as a trade-off between inclusion of the strongest
HDO absorption lines with only minor overlap with the strongest H$_2$O absorption lines. The smaller
spectral windows in respectively blue (H$_2$O) and red (CH$_4$) indicate the weak and strong absorption
bands used for cloud filtering (see below). Although we additionally fit H$_2^{18}$O to improve the fit
quality of the other species, its absorption lines are weaker than those of HDO (not shown). An
accurate retrieval of total column H$_2^{18}$O in this spectral range is not yet feasible and therefore not
part of the final retrieval product.

For the following analysis, we define two relative profiles:

$$\rho_k^{\mathrm{rel}} = \frac{\rho_k}{c_k} \tag{5}$$

and

$$\rho_{k,k'}^{\mathrm{rel}} = \frac{\rho_{k'}}{\rho_k} \,, \tag{6}$$

where $\rho_k^{\mathrm{rel}}$ is the relative profile of absorber $k$ with respect to the vertically integrated total column

$$c_k = \int \rho_k(z)dz \,, \tag{7}$$

and $\rho_{k,k'}^{\mathrm{rel}}$ is the relative profile of absorber $k'$ with respect to absorber $k$. Assuming that the abun-
dance of trace gas $k$ changes by a scaling of the reference profile $\rho_k^{\mathrm{rel}}$, the derivative of the total
optical depth with respect to the trace gas column density is given by

$$\frac{\partial \tau_{\mathrm{tot}}}{\partial c_k} = \frac{1}{c_k} \int \sigma_k(z, \lambda) \, \rho_k(z) \, dz \tag{8}$$





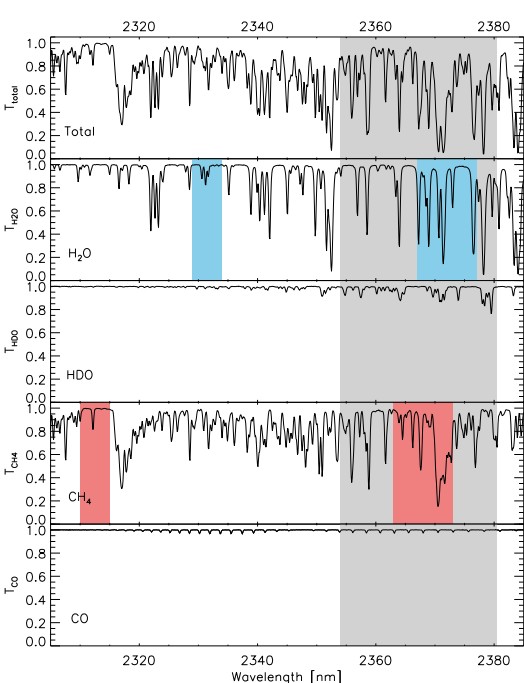

**Fig. 1.** Simulated spectral transmittance in the SWIR spectral range, showing the total transmittance (top panel) as well as the absorption features of the individual species (lower 4 panels). The simulation was performed assuming a solar zenith angle of $0°$ and a viewing zenith angle of $40°$. The 2354.0–2380.5 nm retrieval window is indicated in grey. The coloured windows highlight the weak and strong absorption bands of $H_2O$ (blue) and $CH_4$ (red), used for cloud filtering.

150     and thus

$$\frac{\partial I^{\mathrm{TOA}}}{\partial c_k} = -\frac{I^{\mathrm{TOA}}}{\tilde{\mu} c_k} \int \sigma_k(z) \rho_k(z) dz \,. \tag{9}$$

Corresponding expressions hold for the radiance derivative with respect to the trace gas concentration at a certain altitude level. Finally, the derivative of $I^{\mathrm{TOA}}$ with respect to surface albedo $A_s$ is

155     $$\frac{\partial I^{\mathrm{TOA}}}{\partial A_s} = \frac{\mu_0 F_0}{\pi} \exp\left(-\frac{1}{\tilde{\mu}} \tau_{\mathrm{tot}}\right) \,. \tag{10}$$



After the spectral convolution in equation (1), we have a linearised forward model,

$$\mathbf{F}(\mathbf{x}, \mathbf{b}) = \mathbf{F}(\mathbf{x}_0, \mathbf{b}) + \mathbf{K}\{\mathbf{x} - \mathbf{x}_0\} + \mathcal{O}(\mathbf{x}^2) \qquad (11)$$

with de Jacobian $\mathbf{K} = \frac{\partial \mathbf{F}}{\partial \mathbf{x}}(\mathbf{x}_0, \mathbf{b})$. Here, we distinguish between the state vector $\mathbf{x}$ that comprises in its components the parameters to be retrieved and forward model parameters $\mathbf{b}$ describing parameters other than the state vector that influence the measurement. Equation (11) represents a Taylor expansion of the forward model around state vector $\mathbf{x}_0$ truncated to first order.

### 2.2 Inversion

To determine the column density of water vapour isotopologues from SWIR measurements, we adjust the state vector $\mathbf{x}$ to fit the forward model to the measurement vector $\mathbf{y}$, with spectral residuals $\mathbf{e}_y$, by a least squares fitting approach. The state vector $\mathbf{x}$ includes the total columns of CO, $CH_4$, $H_2O$, $H_2^{18}O$ and HDO, two coefficients to describe the linear spectral dependence of the surfacae albedo $A_s$, and a spectral shift of the ISRF to adjust the spectral calibration of the TROPOMI instrument per retrieval. We apply the profile scaling approach as described by Borsdorff et al. (2014) employing a Gauss-Newton iteration scheme. The least squares minimisation problem

$$\hat{\mathbf{x}} = \min_{\mathbf{x}} ||\mathbf{S}_{\mathbf{y}}^{-1/2}(\mathbf{F}(\mathbf{x}) - \mathbf{y})||^2 \qquad (12)$$

is solved per iteration step with the solution

$$\hat{\mathbf{x}} = \mathbf{x}_0 + \mathbf{G}(\mathbf{y} - \mathbf{F}(\mathbf{x}_0)) \qquad (13)$$

with the gain matrix

$$\mathbf{G} = (\mathbf{K}^T \mathbf{S}_{\mathbf{y}}^{-1} \mathbf{K})^{-1} \mathbf{K}^T \mathbf{S}_{\mathbf{y}}^{-1} \qquad (14)$$

and the measurement covariance matrix $\mathbf{S_y}$.

After convergence, the column averaging kernel can be calculated in a straight forward manner:

$$\mathbf{A}_{k,k'} = \frac{dc_{\mathrm{ret},k}}{d\rho_{k'}} = \mathbf{g}_k \mathbf{K}_{k'}^{\mathrm{prof}} \ . \qquad (15)$$

$\mathbf{g}_k$ is the row vector of the gain matrix $\mathbf{G}$ that belongs to the trace gas $k$ and $\mathbf{K}_{k'}^{\mathrm{prof}}$ is the forward model Jacobian with respect to the trace gas profile $\rho_{k'}$. The height dependence of the profile $\rho_{k'}$ is omitted for a clear presentation. For the full mathematical proof, the reader is referred to Borsdorff et al. (2014). For $k \neq k'$, $\mathbf{A}_{k,k'}$ describes the interference of the retrieved column $c_k$ with the real trace gas vertical distribution of another trace gas $k'$. For $k = k'$, it is the standard column averaging kernel and we use the more simple notation $\mathbf{A}_k = \mathbf{A}_{k,k}$.

The relation between our retrieval product $c_{\mathrm{ret},k}$ (the retrieved total column of species $k$) and the true state (concentration profile $\rho_k$) of the atmosphere is given by

$$c_{\mathrm{ret},k} = \mathbf{A}_k \rho_k + \sum_{k' \neq k} \mathbf{A}_{k,k'} \rho_{k'} + \mathbf{e}_x \ , \qquad (16)$$



where $\mathbf{e}_x$ is the error on the retrieved total column due to the forward model and measurement errors
$\mathbf{e}_y$. In the case of a single trace gas retrieval the interference terms $\mathbf{A}_{k,k'}$ do not exist. In such case,

190 the meaning of the remaining column averaging kernel $\mathbf{A}_k$ is related to the proper choice of the
reference profile and the effective null-space of the regularisation (see the discussion in Borsdorff
et al. (2014) and Wassmann et al. (2015)). If the chosen reference profile is correct, the equation is
equal to a geometrical integration of $\rho_k$. In the case of multiple trace gas retrievals we need to assess
Eq. 16 in more detail. Using Eq. 6 the above equation can be written as

195 
$$c_{\mathrm{ret},k} = \left[ \mathbf{A}_k + \sum_{k' \neq k} \rho_{k,k'}^{\mathrm{rel}} \mathbf{A}_{k,k'} \right] \rho_k + \mathbf{e}_x \, , \tag{17}$$

showing that the contribution of the interference kernel can be interpreted as an error term for every
level of the averaging kernel. Because atmospheric humidity can show strong variability (in both
time and space), and the variability of HDO is (to first order) strongly correlated to the variability of
the main water isotopologue, we are particularly interested in possible interferences between $H_2O$

200 and HDO. We want to be certain that a measured variability in HDO is truly caused by variations
in HDO, and not a result of the interferences between $H_2O$ and HDO. Therefore, we need to test if
the interferences are small for the cases $k = H_2O$ and $k' = HDO$ and vice versa.

  Figure 2 shows an example of the column averaging kernels $\mathbf{A}_{H_2O}$ and $\mathbf{A}_{HDO}$, including the
interference kernels multiplied with the relative profiles as in Eq. 17. The averaging kernel for

205 $H_2O$ ($\mathbf{A}_{H_2O}$, top left panel) shows that the retrieval is only sensitive to $H_2O$ in the lower atmo-
sphere. This is a result of strong pressure broadening of the $H_2O$ absorption lines (Frankenberg
et al., 2009). Since the HDO lines are weaker, the averaging kernel for HDO ($\mathbf{A}_{HDO}$, lower left
panel) is more uniform, showing only slightly lower sensitivity at high layers. The interference
kernels show that above $\sim 10$ km variations of HDO have a minor impact on the retrieval of $H_2O$

210 ($\rho_{H_2O,HDO}^{\mathrm{rel}} \mathbf{A}_{H_2O,HDO} \approx -0.02$, top right panel in Fig. 2), and variations of $H_2O$ have a small im-
pact on the retrieval of HDO ($\rho_{HDO,H_2O}^{\mathrm{rel}} \mathbf{A}_{HDO,H_2O} \approx -0.04 \pm 0.01$, lower right panel in Fig. 2).
Since the column averaging kernels $\mathbf{A}_{H_2O}$ and $\mathbf{A}_{HDO}$ are much larger, and the density profiles $\rho_{H_2O}$
and $\rho_{HDO}$ will be very low higher in the atmosphere, the induced errors on the total columns due to
this interference are practically negligible.

215 For a proper error characterisation of the retrieval product, we calculate the error covariance matrix
$\mathbf{S_x}$ by

$$\mathbf{S_x} = \mathbf{G} \mathbf{S_y} \mathbf{G}^T \, . \tag{18}$$

This allows us to quantify the retrieval noise standard deviation $\sigma_k$ of the individual column densities
and a possible correlation between those.

220 For data interpretation, it is common to consider the relative abundance of HDO with respect to
$H_2O$: $r = c_{\mathrm{ret,HDO}} / c_{\mathrm{ret},H_2O}$, and to reference the ratio to the Vienna Standard Mean Ocean Water



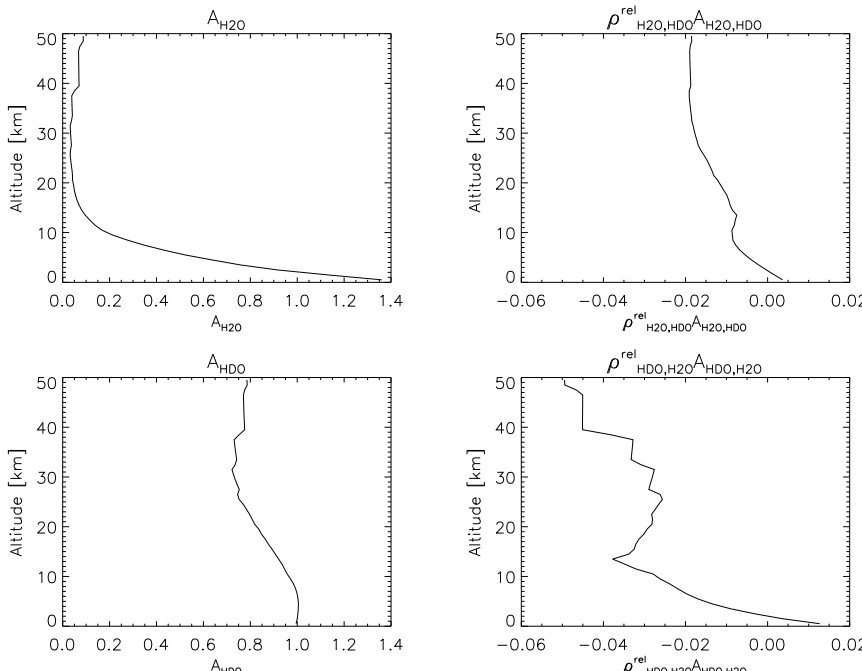

**Fig. 2.** Top left: total column averaging kernel for $H_2O$. Lower left: total column averaging kernel for HDO. Top right: the sensitivity of total column $H_2O$ to variations in HDO at different altitudes. Lower right: the sensitivity of total column HDO to variations in $H_2O$ at different altitudes.

(VSMOW) ratio $r_s = 3.1153 \cdot 10^{-4}$:

$$\delta D = \left[ \frac{r}{r_s} - 1 \right],$$

(19)

where $\delta D$ is typically given in units of per mil. The first studies that measured $\delta D$ in the atmosphere
225   (as mentioned in the introduction) have shown that typical variations in $\delta D$ over time or space are of
the order of 50–100‰, which we therefore regard as the required accuracy for a useful product. The
diagnostic tools of the individual columns can be used to derive the corresponding quantities for $\delta D$.
For example, the standard deviation of the retrieval noise is given by

$$\sigma_{\delta D} = \frac{r}{r_s} \sqrt{\frac{\sigma_{H_2O}^2}{c_{H_2O}^2} + \frac{\sigma_{HDO}^2}{c_{HDO}^2} - \frac{2S_{HDO,H_2O}}{c_{H_2O} c_{HDO}}}$$

(20)

230   and in a similar manner the column averaging kernel with respect to the $H_2O$ and HDO abundance
can be derived.





### 2.3 Two-band cloud filtering

Since the HDO and $H_2O$ retrieval algorithm does not account for clouds nor any other scattering layers, we need to filter for clouds to avoid large retrieval inaccuracies. This filtering is achieved using the retrieved columns in a weak and strong absorption band of either $CH_4$ or $H_2O$. The bands used are indicated in Fig. 1. Elevated scattering layers not accounted for in the forward model generally cause a retrieval bias by scattering photons directly into the instrument. This optical path length shortening leads to negative biases in the retrieved total columns. The two-band cloud filter relies on the fact that a total column measurement using a strong absorption band is more strongly affected by this "shielding bias" than the measurement using a weak absorption band. As a result, the relative difference in the retrieved total column between the weak and strong absorption band can be used to indicate the presence of clouds. Using a set of simulated measurements for varying cloudy conditions (as will be described in Sect. 3.1), we have tested that using a threshold $< 6\%$ for the relative difference in total column $CH_4$ between the weak and strong bands we filter for ground scenes that have a cloud fraction of more than 10–20% (cloud top height $\geq 1$ km). Scenes with low-level clouds (cloud top heights $< 1$ km) are not affected by this filter. Since low-level clouds above sea pass the filter, the retrieval allows for measurements over sea above these clouds due to their high albedo. The albedo of the sea surface itself is too low in the shortwave infrared for a meaningful retrieval above cloud-free sea pixels.

As an example, Fig. 3 shows how the relative difference in retrieved methane absorption between the strong and weak absorption bands (shown in red in Fig. 1) changes for the scenarios with clouds (left panel), cirrus (middle panel) and aerosol (right panel). Scenes with strongest effects on the light path of the observed signal will also show the largest relative difference. We find that with a relative difference in methane absorption $< 6\%$ we effectively filter for clouds and cirrus, as well as for low surface albedo scenes affected by aerosol. For example, not affected by the filter are scenes with a cloud top height $\lesssim 1$ km or scenes with a low fraction of higher-level clouds (i.e. the contours with the two darkest shades of blue in the left panel of Fig. 3). A similar performance is achieved with a two-band water filter, using the weak and strong water bands as shown in blue in Fig. 1 and a threshold for the relative difference in water absorption in these bands of 8%. In the next sections the impact of this cloud filter on the retrievals of HDO and $H_2O$ will be shown.

The two-band cloud filtering will be part of TROPOMI's operational methane pre-processing pipeline (Hu et al., 2016), and so synergies with the operational data processing can be used to reduce the processing time significanty, as we have estimated that on average 20% of all the measured ground scenes will pass the cloud filter above land, and 14% above sea.



**Table 1.** Overview of the generic scenarios used for the performance analysis. For the scenarios where the SZA was not variable, it was fixed at $50°$. For the clouds scenario the surface albedo was fixed at 0.05.

| Scenario | Variable X | Variable Y |
|---|---|---|
| cloud free | surface albedo: 0.03–0.6 | SZA: 0.0–70.0$°$ |
| clouds | cloud top height: 1–8 km | cloud fraction: 0.0–1.0 |
| cirrus | surface albedo: 0.03–0.6 | $\tau_{\mathrm{cir}}$ (2300 nm): 0.0–1.0 |
| aerosol | surface albedo: 0.03–0.6 | AOT (550 nm): 0.0–1.0 |

## 3 Performance analysis for generic scenarios

To assess the performance of the retrieval algorithm, we applied the retrieval to simulated measurements for various generic scenarios. For each scenario, we systematically varied two variables such as surface albedo, solar zenith angle (SZA), cloud parameters (cloud top height, cloud fraction and cloud optical thickness ($\tau_{\mathrm{cld}}$)) and aerosol optical thickness (AOT). An overview of the scenarios is given in Table 1.

### 3.1 Measurement simulations

The measurement simulations for the generic scenarios were created using the S-LINTRAN radiative transfer model (Schepers et al., 2014). The implementation of S-LINTRAN for TROPOMI simulations, including the instrument model, is described in detail in Landgraf et al. (2016), as the same simulations have been used to assess the performance of the CO retrieval algorithm. A summary of the implementation is provided in the following two paragraphs.

The model is a scalar plane-parallel radiative transfer model that fully accounts for multiple elastic light scattering by clouds, cirrus, air molecules and the reflection of light by the Earth surface. The optical properties of water clouds are calculated using Mie theory with microphysical cloud properties given in Table 2. For ice clouds the ray tracing model of Hess and Wiegner (1994); Hess et al. (1998) is employed assuming hexagonal, columnar ice crystals randomly oriented in space. Cirrus and water clouds are described by a cloud top and base height, and a cloud optical thickness. While cirrus fully cover the observed ground scene, water clouds can show partial cloud coverage by utilising the independent pixel approximation (Marshak et al., 1995) for the simulation.

Measurement noise was superimposed on the radiance spectra using the TROPOMI noise model (Tol et al., 2011). This assumed an observed ground scene of $7 \times 7$ km$^2$ and a telescope aperture of $6 \cdot 10^{-6}$ m$^2$. The resulting signal to noise ratio is 120 in the continuum of the spectrum for a dark reference scene (surface albedo $A_{\mathrm{s}} = 0.05$, viewing zenith angle VZA $= 0°$ and solar zenith angle SZA $= 70°$).

The atmospheric model assumed the US standard atmosphere (1976) for the profiles of dry air





**Table 2.** Microphysical properties of water and ice clouds: $n(r)$ represents the size distribution type, $r_{\text{eff}}$ and $v_{\text{eff}}$ are the effective radius and variance of the size distribution, $m = n - ik$ is the refractive index. The ice cloud size distribution follows a power-law distribution as proposed by Heymsfield and Platt (1984).

|  | water clouds | ice clouds |
| --- | --- | --- |
| $n(r)$ | gamma | $(r/r_i)^{-3.85}$ |
| $r_{\text{eff}}$ [μm] | 20 | - |
| $v_{\text{eff}}$ | 0.10 | - |
| $n$ | 1.28 | 1.26 |
| $k$ | $4.7 \cdot 10^{-4}$ | $2.87 \cdot 10^{-4}$ |

density, temperature, pressure, water and CO. The $CH_4$ profile is taken from the CAMELOT European background profile scenario (Levelt and Veefkind, 2009), interpolated to the same pressure grid and converted from mixing ratios to densities using the air densities from the US standard atmosphere. We separated the water profile into individual profiles for the three isotopic components

with absorption features in the TROPOMI SWIR range: $H_2^{16}O$, $H_2^{18}O$ and HDO. First, the water profile was scaled with the VSMOW abundance of the respective species. Additionally, a realistic altitude-dependent depletion of HDO and $H_2^{18}O$ was assumed. For HDO we assumed a linear decrease from $\delta D = -100\,‰$ at the surface to $\delta D = -600\,‰$ at 15 km, followed by a linear increase to $\delta D = -400\,‰$ at the top of the atmosphere at an altitude of 48 km (Ehhalt, 1974; Ehhalt et al.,

2005; Schneider et al., 2010). We further assumed that the concentration of $H_2^{18}O$ is related to the concentration of HDO according to the empirically determined "global meteoric water line" (Craig, 1961)

$$\delta D = 8 \cdot \delta^{18}O + 10‰ , \tag{21}$$

where $\delta^{18}O$ is defined in the same way as $\delta D$ (Eq. 19). All the atmospheric profiles used for the

measurement simulations are shown in Fig. 4.

In the following subsections, we characterise the retrieval performance for the generic scenarios considering separately the retrieval statistical errors (i.e. the single measurement noise) $\sigma_{H_2O}$, $\sigma_{HDO}$ and $\sigma_{\delta D}$ (neglecting the small HDO–$H_2O$ cross-correlation term for $\sigma_{\delta D}$) and the biases in the total columns $c_{\text{ret,H}_2\text{O}}$, $c_{\text{ret,HDO}}$ and their ratio $\delta D$. The retrieval statistical error estimate for $\delta D$ is

given by Eq. 20. For the bias in $\delta D$, which we refer to as "$\Delta \delta D$", we first determine $\delta D_{\text{retrieval}}$ by removing the noise on the retrieved total columns HDO and $H_2^{16}O$ using linear error propagation for the particular noise realisation. We need to compare $\delta D_{\text{retrieval}}$ with $\delta D_{\text{model}}$, where $\delta D_{\text{model}}$ is $\delta D$



of the "true" model atmosphere:

$$\delta D_{model} = \frac{c_{true,HDO}}{c_{true,H_2O}} \frac{1}{r_s} - 1 .$$ (22)

Finally, the retrieval bias on $\delta D$ is defined as:

$$\Delta \delta D = \delta D_{retrieval} - \delta D_{model} .$$ (23)

### 3.2 Cloud-free conditions

In Fig. 5 we show the simulated cloud-free retrieval bias for the total column $H_2O$ (left panel), total column HDO (middle panel) and their ratio ($\Delta \delta D$, right panel), as a function of surface albedo and
SZA (no clouds or aerosol present). The figure shows that the retrieval performs very well for the majority of the scenes, with $\Delta \delta D$ less than 0.8‰. Only for the lowest surface albedos (0.03–0.05) the bias in $\delta D$ increases to a few per mil, due to slightly more negative bias in $H_2O$ compared to HDO.

    The corresponding statistical error estimates are shown in Fig. 6. $\sigma_{H_2O}$ reaches maximum values
of 1.6–2.0% for the lowest surface albedos and highest SZAs and $\sigma_{HDO}$ is about a factor 2 larger due to the weaker HDO absorption features. Combined, it results in values for $\sigma_{\delta D}$ of the order of 15–25‰ for the lowest surface albedos. For high surface albedo regions such as deserts (surface albedo $\sim 0.3$ in the SWIR) typical values for $\sigma_{\delta D}$ are 2–4‰. This is roughly an order of magnitude better than what is achieved with SCIAMACHY (Scheepmaker et al., 2015).

### 3.3 Clouds and cirrus

As the retrieval algorithm does not account for scattering, any clouds, cirrus and aerosol present in the observed scene will lead to biases in the retrieval of the total columns HDO and $H_2O$. We have tested the performance of the retrieval under cloudy conditions with a scenario assuming a cloud with an optical thickness of $\tau_{cld} = 5$, with varying cloud top heights (between 1–8 km in steps of
1 km) and varying cloud fractions (between 0.0–1.0 in steps of 0.1). The same scenario was used to demonstrate the two-band cloud filter in the left panel of Fig. 3. Due to differences in their retrieval sensitivities, the observed bias is stronger for $H_2O$ than for HDO, leading to significant biases in their ratio, increasing with both cloud fraction and cloud top height (although not shown, we find that $\Delta \delta D$ can reach values $> 900$‰ for clouds above 7 km with 100% cloud coverage). Similarly,
by simulating scenarios with varying surface albedos and a uniform cirrus or aerosol layer with varying optical thickness (see Table 1), we find that this bias increases with the optical thickness of the layer and with lower surface albedos, as both lead to a lower contribution of photons from below the scattering layer reaching the instrument. As described in Sect. 2.3, the two-band cloud filtering technique will be used to pre-filter the scenes most affected by this shielding bias. We find that, after





applying the two-band methane filter to scenes affected by clouds and cirrus, $\Delta \delta D \lesssim 70‰$ and
$\sigma_{\delta D} = 10\text{–}20‰$ (for scenes with a low surface albedo of $A_s = 0.05$).

### 3.4 Aerosol

An aerosol layer will typically have a lower optical thickness than clouds, and occurs lower in
the atmosphere, leading to a different impact on our non-scattering retrieval. Our aerosol scenario
assumes a uniform layer of a sulphate-type aerosol in the boundary layer between 0–2 km. Figure 7
shows how this induces a bias in the total columns $H_2O$, HDO and their ratio, as a function of aerosol
optical thickness and surface albedo. We see that for very low surface albedos, direct reflection off
the aerosol layer leads to path length shortening and a corresponding negative (shielding) bias for
the total column $H_2O$. This effect is weaker for HDO due its more uniform averaging kernel. For
higher surface albedos, however, we see that the bias becomes positive, likely due to an increased
amount of light scattering in the boundary layer. The contribution of photons from the brighter
surface increases and a fraction of these photons undergo multiple scattering events between the
aerosol layer and the surface, enhancing the path length. The net effect on $\delta D$ is that its bias due
to aerosol is highest for the lowest surface albedos and highest AOT (right panel in Fig. 7). If we
take the two-band cloud filter into account (right panel in Fig. 3) to filter the lowest surface albedos
affected by aerosol, we are left with $\Delta \delta D \lesssim 20‰$ due to boundary layer aerosol with AOT = 1.0 (at
550 nm). The statistical error (not shown) does not depend significantly on AOT, but varies primarily
with surface albedo, reaching similar peak values as in the cloud free scenario ($\sigma_{\delta D} \approx 20‰$).

### 3.5 Summary of the general performance

In summary, we can conclude that the retrieval performs well under cloud-free conditions. The bias
$\Delta \delta D$ will be less than 2‰, even for the lowest surface albedos, and the statistical errors vary from
2–4‰ for high albedos to 15–25‰ for the lowest albedos. Under conditions with clouds, cirrus or
aerosol the retrieval performs less well and we generally find a positive bias in $\delta D$. To restrict this
bias we need strict filtering against clouds and aerosol by applying the two-band cloud filter either
to methane or water (which additionally leads to a great reduction in the computational effort).
Applying a two-band methane threshold of 6%, we restrict the bias in $\delta D$ to $\Delta \delta D < 70‰$ for all
simulated measurements. Averaging multiple single measurements over time and space will further
reduce the statistical error and will improve the accuracy to better than the maximum 70‰, bringing
the measurements within the requirements to study typical temporal and spatial gradients (which are
of the order of 50–100‰).





**Table 3.** Summary of the sensitivity to the meteorological input and instrument parameters, expressed as the mean difference in $\delta$D between the perturbed and default retrievals for the 45 scenes with the three lowest surface albedos (0.03, 0.05 and 0.075).

| Prior parameter | Systematic error in $\delta$D [‰] | | |
| --- | --- | --- | --- |
| Temp $-0.5$ K | $-6.9$ | $\pm$ | 0.74 |
| Temp $+0.5$ K | $+7.0$ | $\pm$ | 0.11 |
| Temp $-1$ K | $-14$ | $\pm$ | 0.74 |
| Temp $+1$ K | $+14$ | $\pm$ | 0.17 |
| Pressure $\times 0.99$ | $-4.5$ | $\pm$ | 0.74 |
| Pressure $\times 1.01$ | $+4.5$ | $\pm$ | 0.54 |
| Rad. offset $+0.1\%$ | $-0.054$ | $\pm$ | 0.11 |
| Rad. offset $+0.5\%$ | $-0.20$ | $\pm$ | 0.16 |
| ISRF FWHM $-1\%$ | $+0.36$ | $\pm$ | 0.85 |
| ISRF FWHM $+1\%$ | $-0.33$ | $\pm$ | 0.84 |

## 4 Sensitivity to prior assumptions

Similarly to what was done for the CO TROPOMI retrievals (Landgraf et al., 2016), we have tested the sensitivity of the $H_2O$ and HDO total column retrievals to the prior assumptions, including the impact on $\delta$D. These so-called forward modelling errors were tested on the cloud-free scenario

(with varying SZA and surface albedo) using the same measurement simulation as described in Sect. 3. A perturbation in one of the input assumptions was introduced, after which the retrievals were performed and compared with the default retrievals without the perturbation. The impact of the perturbation is expressed as a systematic error and standard deviation, where we define the systematic error as the mean difference in $\delta$D between the perturbed and default retrievals for the 45

scenes with the three lowest surface albedos (0.03, 0.05 and 0.075). The results are summarised in Table 3.

The retrieval uses a priori temperature profiles and surface pressures from the ECMWF. To test the impact of uncertainties in the temperature profile, we have varied this profile by $\pm 0.5$ K and $\pm 1$ K. This is primarily affecting the retrieved total column $H_2O$, while the total column HDO is not very

sensitive to temperature variations. A perturbation of $+1$ K ($-1$ K) leads to a decrease (increase) in the retrieved $H_2O$ column of 1.8%, inducing a systematic error in $\delta$D of $+14$‰ ($-14$‰). This error is constant for all surface albedos and SZAs and scales linearly with the size of the temperature perturbation.

The atmospheric pressure profile is derived from the surface pressure. To test the impact on



inaccuracies in the ECMWF surface pressure, we applied a perturbation of $\pm 1\%$. This leads to systematic errors of about 0.5% in $H_2O$ and 0.13% in HDO (with reversed sign), together inducing errors of about 4.5‰ in $\delta$D.

The retrieval algorithm requires a reflectance spectrum, acquired by dividing the radiance spectrum measured from the Earth's surface by the irradiance spectrum measured directly from the Sun.
Differences in the radiometric offset between these spectra could induce spectral features in the reflectance spectrum, leading to systematic errors. The TROPOMI instrument requirement for the radiometric offset on the radiance is 0.1% of the continuum level. We have tested the impact of an offset on the radiance of 0.1% and 0.5% of the maximum value in the retrieval window. However, the retrieval fits for an offset in the reflectance spectra, which partly mitigates the effects of an offset
in the radiance or irradiance. The systematic errors due to uncertainties in the radiometric offset are therefore very small (errors in $\delta$D less than 0.5‰).

For the default retrieval and measurement simulations we have assumed a Gaussian slit function (ISRF) with a FWHM of 0.25 nm. We have tested the impact of perturbing this FWHM by $\pm 1\%$ and find that the induced systematic errors are strongly dependent on surface albedo and SZA. The
largest errors in $\delta$D reach $\pm 3$‰ and are found for high albedos and low SZAs. The mean systematic error for the lowest albedos is $0.36 \pm 0.85$‰.

In summary (also see Table 3), we find that the retrieval algorithm is most sensitive to uncertainties in the a priori temperature profiles, followed by the pressure profiles. The sensitivity to uncertainties in the instrument parameters is about an order of magnitude smaller. The uncertainties in the input
profiles are expected to be mostly quasi-random in nature, which means their impact on the error in $\delta$D will diminish when taking averages in time and space.

More structural systematic errors (i.e. those that will not diminish by averaging) are potentially caused by uncertainties in the water spectroscopy. Recent studies have shown that spectroscopic uncertainties of water can have a large impact on total column retrievals of CO (Galli et al., 2012), $CH_4$
(Frankenberg et al., 2008; Schneising et al., 2009), $H_2O$ (Schrijver et al., 2009) and the HDO/$H_2O$ ratio (Scheepmaker et al., 2013). As a test of the possible impact of uncertainties in the water line parameters, we have repeated the retrievals of the simulated clear-sky scenario after replacing the line parameters of the water isotopologues. For the simulated spectra the parameters from HITRAN (Rothman et al., 2009) were used. We then performed the retrievals using the water line parame-
ters from Scheepmaker et al. (2013). Table 4 shows the induced systematic errors for replacing a single isotopologue at a time, and for replacing all modelled water isotopologues simultaneously. It shows that the retrieval of HDO and $H_2O$ can be very sensitive to spectroscopic uncertainties, especially in the ratio HDO/$H_2O$, since HDO and $H_2^{16}O$ can show sensitivities with opposite sign, which strengthen each other when taking the ratio (as can be seen from replacing only the $H_2^{16}O$
parameters). The differences in spectroscopy between HITRAN and Scheepmaker et al. (2013) can lead to differences in $\delta$D of up to 128‰. Although we find that the differences do not depend on



**Table 4.** Sensitivity to a change of water line parameters from HITRAN2008 to the parameters from Scheepmaker et al. (2013).

| Isotopologue | Systematic Error | | | |
|---|---|---|---|---|
| | $H_2^{16}O$ [%] | HDO [%] | $CH_4$ [%] | $\delta D$ [‰] |
| $H_2^{16}O$ | $+8.0 \pm 0.55$ | $-4.6 \pm 0.50$ | $+1.3 \pm 0.35$ | $-97 \pm 1.5$ |
| HDO | $-0.069 \pm 0.036$ | $-4.2 \pm 0.043$ | $-0.040 \pm 0.037$ | $-34 \pm 0.29$ |
| $H_2^{18}O$ | $+0.011 \pm 0.017$ | $-0.11 \pm 0.0090$ | $-0.029 \pm 0.017$ | $-0.97 \pm 0.21$ |
| All | $+7.9 \pm 0.53$ | $-8.7 \pm 0.50$ | $+1.3 \pm 0.37$ | $-128 \pm 1.5$ |

surface albedo or SZA, we cannot exclude a dependency on the total amount of water vapour, that might lead to seasonal and latitudinal biases. Similar to the retrieval of CO (Galli et al., 2012), the HDO/$H_2O$ retrieval will very likely benefit from a reassessment of the spectroscopic line parameters

of water, a study which is currently on-going (Loos et al., 2015). Regardless of such reassessments, validation studies will be needed to verify spectroscopy and to define corrections that might mitigate spectroscopy related biases.

## 5 Performance analysis for a realistic scenario

To show the capabilities of the TROPOMI $H_2O$ and HDO total column retrievals, we have sim-

ulated an ensemble of measurements that reflect a realistic scenario as accurately as possible. In Sect. 5.1 we describe the input data and measurement simulations in more detail. In Sect. 5.2 we discuss the results of retrieving the simulated measurements, in terms of retrieval bias, precision, and effectiveness of the cloud filtering.

### 5.1 Measurement simulations

The simulated measurement ensemble covers a region over the South-West part of the US and the North-East part of Mexico as TROPOMI would observe it on August 4[th] 2009. Figures 8 and 9 show the region in terms of various input fields. This region comprises a clear gradient in the relative abundance of HDO with respect to $H_2O$ due to the transport of humid air from coastal regions inland. We have combined data from the MODIS Aqua satellite (clouds, land/water coverage, sur-

face albedo, aerosol) with data from ETOPO5 (elevation), ECMWF (surface pressure, temperature profiles, specific humidity), TM5 (CO and $CH_4$ profiles) and LMDZiso (HDO and $H_2^{18}O$ profiles) to simulate 27405 TROPOMI measurements on a grid of 135 by 203 ground pixels. This ensemble represents roughly what TROPOMI will observe in five minutes with a daily revisiting cycle.

The viewing and solar geometry and ground pixel size were adapted from MODIS Aqua granule

2009216 (19h45m UT), where the MODIS information was spatially resampled on a pixel size of



$10 \times 10$ km$^2$ at sub-satellite point. The pixel distortion towards the outer swath was adopted from the MODIS observation. The surface reflection was estimated from the MODIS MCD43C4 data product at 2105–2155 nm for the same period (Strahler et al., 1999) in combination with the surface elevation from ETOPO5 (NOAA, 1988). Furthermore, the MODIS MYD06 cloud product (Platnick et al., 2015) was used to estimate cloud cover and cloud top height for the individual TROPOMI ground pixels. Only clouds with top height above 100 meters were used. We derived cirrus optical thicknesses from the MODIS cirrus reflectance product employing the algorithm by Dessler and Yang (2003). For all pixels, the cirrus was located between 9–10 km. For the aerosol optical thickness (at 550 nm) we used the MODIS MYD08_M3 global monthly mean product (Platnick, 2015), resampled to the above mentioned granule with a pixel size of $10 \times 10$ km$^2$ at sub-satellite point. For some pixels with missing aerosol data the optical thickness was set to 0.1. We assumed three different aerosol types: "oceanic" above water, "dust" above land and "urban" above all land regions with AOT $> 0.23$. The corresponding model fields are depicted in Fig. 8.

The distribution of atmospheric trace gases was estimated using TM5 chemistry model simulations (Krol et al., 2005), which yields the CO and CH$_4$ abundances. Moreover, we used data from ECMWF (Dee et al., 2011) for the atmospheric pressure, temperature and humidity profiles. For realistic HDO/H$_2$O ratios we derived $\delta$D profiles from LMDZiso model simulations (Risi et al., 2010) and for the $\delta^{18}$O profiles we assumed correlation to $\delta$D according the global meteoric water line (also see Eq. 21, Craig, 1961). Figure 9 shows the resulting total columns for the most important species for our ensemble. Based on this input and the TROPOMI instrument model as described in Sect. 3.1, we simulated for each individual pixel the TROPOMI SWIR observations using the S-LINTRAN radiative transfer model.

### 5.2 Results

Using the simulated measurement ensemble we retrieved the water vapour abundances using the SICOR retrieval algorithm including the retrieval of the two methane and water bands used for cloud filtering. In Fig. 10 the results are shown in terms of the retrieved bias in total column H$_2$O, HDO and $\delta$D. We also show the relative difference in the weak vs strong water bands that was used for cloud filtering. The cloud filter panel (lower-left panel in Fig. 10) shows that the algorithm retrieved some pixels above the Gulf of Mexico, even though these pixels did not contain low-level clouds. In reality such pixels will be removed by pre-filtering for very low albedo regions. Once the two-band cloud filter threshold is applied (keeping only pixels with a relative difference $< 8\%$ using the water bands), practically all ocean pixels are removed, as well as all the lands pixels affected by clouds, resulting in 54.5% of the pixels remaining for further study. Using the two methane bands as a cloud filter with a threshold of 6% resulted in slightly less strict filtering (60.7% of the pixels remaining), as certain pixels with low and optical thin clouds in the west above Mexico and in the east above Alabama were not removed (not shown). The few rejected pixels in the centre and south of Texas





show that the cloud filter effectively removed high isolated clouds with low optical thickness (cf. the lower two panels in Fig. 8 with Fig. 10), but left pixels with clouds $< 1$ km intact (as is preferred). The large group of pixels in the north-east of the ensemble were rejected based on the presence of
high and optically thick clouds, or the presence of aerosol above low surface albedo regions.

The other three panels in Fig. 10 show the remaining biases in total column $H_2O$, HDO and $\delta$D after cloud and ocean filtering. Both $H_2O$ and HDO show a positive bias of a few percent above the higher surface albedo regions in the west and a small negative bias over the lower surface albedo regions in the east. Careful inspection of the states of Louisiana and Mississippi show that even the
albedo contrast caused by the Mississippi and Red River basins can be observed in the $H_2O$ and HDO bias maps. We also see that the biases are slightly larger for $H_2O$, compared to HDO. The cause for these biases is aerosol, as the aerosol bias shows the same patterns as a function of surface albedo and AOT, and the effect is slightly larger for $H_2O$, as discussed in Sect. 3.4 and shown in Fig. 7. Combined into a ratio, the lower-right panel in Fig. 10 show that the retrieval bias in $\delta$D
is slightly negative above the highest albedos and increases to a positive bias with a maximum of $\sim 20$‰ above the lowest albedo regions. Areas at high altitudes usually have a lower humidity and therefore a lower $\delta$D compared to areas at lower altitudes. This gradient is visible in the bottom right panel in Fig. 9. Furthermore, areas at higher altitude generally have a higher surface albedo. Because the retrieval bias in $\delta$D is negative for high surface albedos and positive for low surface
albedos, the altitude gradient in $\delta$D is overestimated by the retrieval. This will likely be the case for all scenarios with a gradient between higher elevated (or drier) areas and lower elevated (or more humid) areas.

Figure 11 shows the same maps in terms of single measurement precision error ($1\sigma$). As expected, the dominant factor to determine the precision error is surface albedo. The error of the strong ab-
sorbers $H_2O$ and $CH_4$ is 0.05–0.15% for the highest surface albedos, and increases to 0.35% for the lowest surface albedos. The precision errors of the weak absorber HDO are larger, reaching 0.50% for the lowest surface albedos. This translates into precision errors in $\delta$D of at most 5‰ above the lowest albedos.

This realistic scenario demonstrates the capabilities of TROPOMI and the SICOR algorithm to
retrieve accurate patterns in total column $H_2O$, HDO and $\delta$D above land from a single overpass. After the two-band cloud filter effectively removed all measurements above water and high clouds, a small bias remains due to aerosol, which correlates with surface albedo. The bias is smaller (and of opposite sign) compared to the temporal or spatial gradients in $\delta$D expected for typical science cases (e.g., as observed by SCIAMACHY in Yoshimura et al., 2011; Lee et al., 2012; Risi et al.,
2012a; Okazaki et al., 2015). This ensures the ability to detect and study patterns in $\delta$D on much smaller timescales and at higher spatial resolution compared to previous satellite missions, but it should also be kept in mind to be careful when using the data over regions with strong gradients in surface albedo.



## 6 Discussion and conclusions

We have presented an algorithm and performance analysis for the retrieval of total column $H_2O$
and HDO from TROPOMI measurements onboard the Sentinel-5 Precursor mission. By adapting
ESA's operational CO algorithm (Landgraf et al., 2016), we developed a relatively simple approach
that is fast but relies on strict filtering for clouds, cirrus and aerosol using a two-band methane or
water retrieval. The ratio $HDO/H_2O$ will be a useful scientific product in the fields of hydrology

and climate research, with the potential to improve our understanding of the processes controlling
atmospheric humidity and transport.

The first studies in these directions that used a similar type of column-averaged satellite product
were using SCIAMACHY data (Frankenberg et al., 2009; Yoshimura et al., 2011; Lee et al., 2012;
Risi et al., 2012b,a; Scheepmaker et al., 2015). These studies showed that the typical seasonal or

spatial gradients in $\delta D$ are about 50–100‰. The measurement precision and accuracy needs to be
higher than this in order to contribute significantly to the science. For SCIAMACHY, this implied
either taking monthly averages or binning to a spatial resolution of at least $1 \times 1°$ in order to bring the
statistical error down to about 20‰ (the single measurement precision being $\sim 115$‰, Scheepmaker
et al., 2015). The newer GOSAT measurements show an improvement in precision by a factor

of about two, compared to SCIAMACHY (Frankenberg et al., 2013; Boesch et al., 2013). Both
SCIAMACHY and GOSAT products show a negative bias of about 30–70‰ compared to ground-
based Fourier-transform spectroscopy (FTS) networks.

Our analysis has shown that TROPOMI is expected to deliver a much better performance than
SCIAMACHY and GOSAT in terms of $\delta D$ in only a single overpass. The single measurement noise

will be better than 15–25‰ for even the lowest surface albedos, while at the same time the spatial
resolution of $7 \times 7$ km$^2$ is much higher than SCIAMACHY's $120 \times 30$ km$^2$ and provides a better
coverage than GOSAT's sparse spatial sampling. Even though we still need to filter for clouds,
due to this higher spatial resolution TROPOMI will observe many more useful scenes in between
clouds compared to SCIAMACHY or GOSAT. This allows for new opportunities of studying the

hydrological cycle on timescales of mere days or weeks instead of seasons or years, or over longer
periods if a high spatial resolution is desired.

Mainly due to the presence of low-level aerosol in the atmosphere, the cloud-filtered TROPOMI
measurements of total column HDO and $H_2O$ are not expected to be completely bias free. Changes
to the light paths of the reflected photons due to any scattering particles remaining after filtering are

not accounted for in the retrieval algorithm, and lead to biases of a few percent in total column HDO
and $H_2O$, and up to $\sim 20$‰ in $\delta D$, depending on surface albedo, as shown by our simulated scenario
of measurements above the USA and Mexico.

After launch and commissioning of the instrument in Q4 2016, validation using ground-based FTS
data from the TCCON and NDACC networks is needed to test the performance of the algorithm on

real measurements. Ideally, the $HDO/H_2O$ products from these networks should first be intercom-



pared, both using the results from the on-going reassessment of the water spectroscopy (Loos et al., 2015), and for a range of atmospheric conditions and geographical locations. Any possible differences due to either spectroscopy or location (e.g. as found by Scheepmaker et al., 2015) need to be understood before the next generation of HDO and $H_2O$ global retrievals from space can be ex-

ploited to come to a better understanding of the atmospheric hydrological cycle and the role it plays in our changing climate.

*Acknowledgements.* LMDZiso simulations used the computing resources of IDRIS under the allocation 0292 made by GENCI.



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





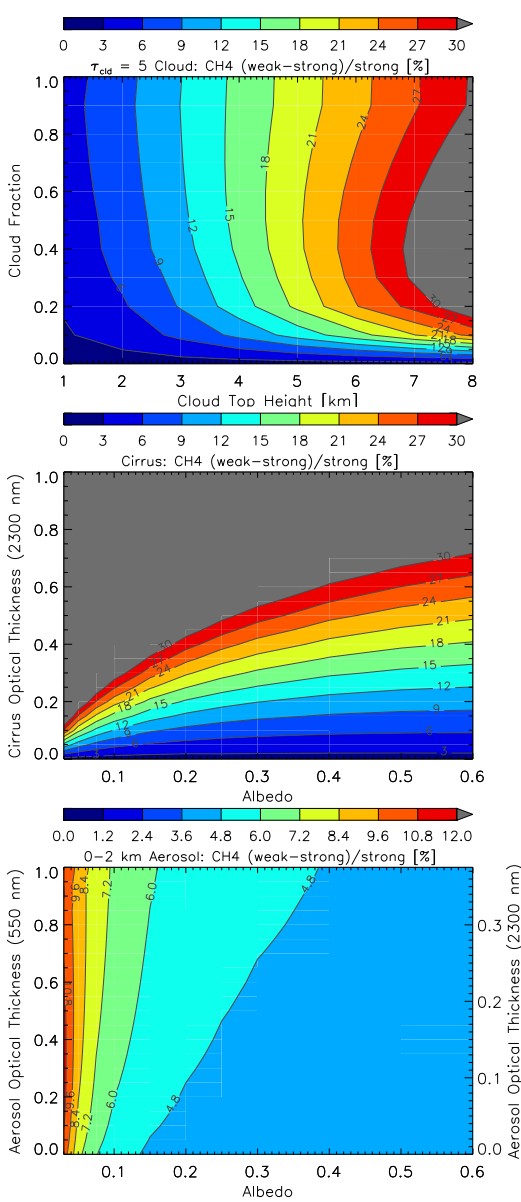

**Fig. 3.** Two-band $CH_4$ filter results for clouds (top), cirrus (middle) and aerosol (bottom). Plotted is the relative difference in total column $CH_4$ retrieved from the weak and strong bands: $CH_4$ (weak - strong) / strong (%). The cloud scenario assumed a cloud optical thickness of $\tau_{cld} = 5$ and a variable cloud-top-height (x-axis) and cloud fraction (y-axis). The cirrus scenario assumed a cloud fraction of 100% for a layer between 9 and 10 km and a variable surface albedo (x-axis) and cirrus optical thickness (y-axis). The aerosol scenario assumed a sulphate-type aerosol in the boundary layer between 0–2 km and a variable surface albedo (x-axis) and aerosol optical thickness (y-axis).





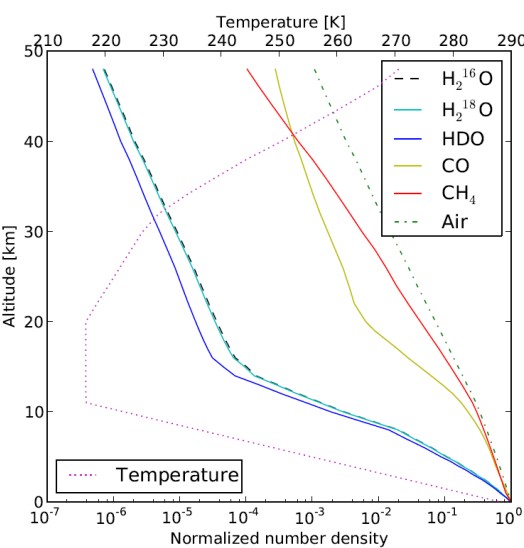

**Fig. 4.** Atmospheric profiles for the number densities of the absorbers (bottom axis, normalised to the surface value) and temperature (top axis) used as input for the model atmosphere.





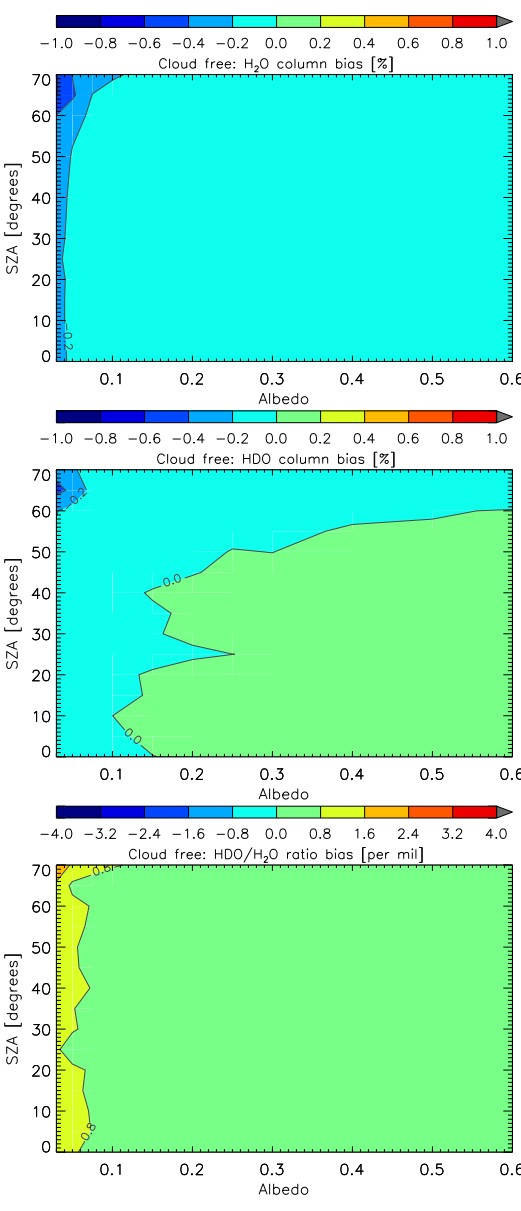

**Fig. 5.** Cloud free rerieval bias as a function of surface albedo and SZA for the total columns of $H_2O$ (top), HDO (middle) and the HDO/$H_2O$ ratio (bottom).





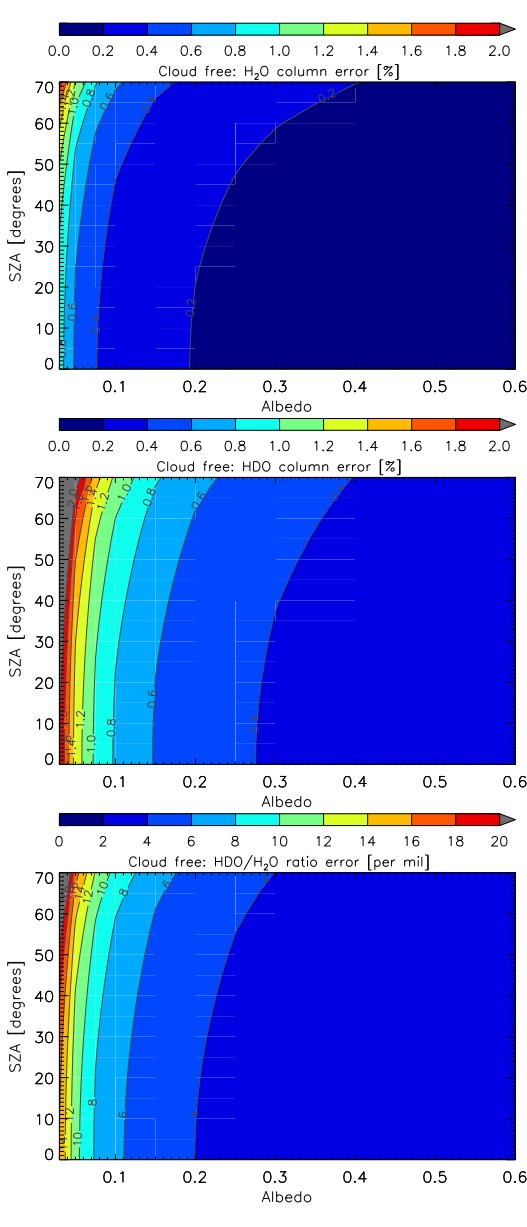

**Fig. 6.** Cloud free statistical error estimates (single measurement noise) as a function of surface albedo and SZA for the total columns of $H_2O$ (top), HDO (middle) and the HDO/$H_2O$ ratio (bottom).





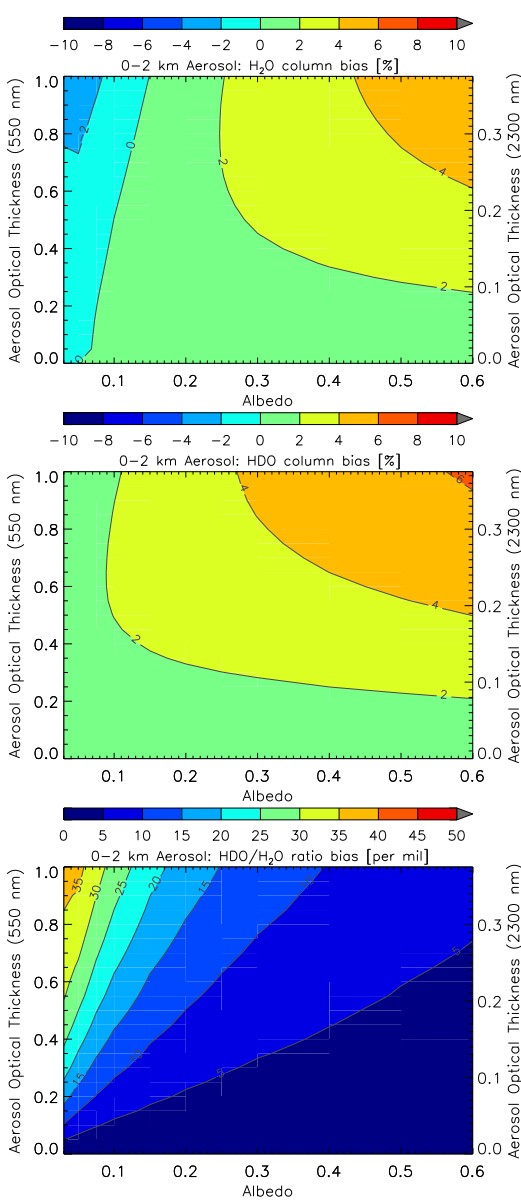

**Fig. 7.** Rerieval bias for an aerosol layer between 0–2 km as a function of surface albedo and AOT for the total columns of $H_2O$ (%) (top), HDO (%) (middle) and the HDO/$H_2O$ ratio (‰) (bottom). Applying the two-band methane cloud filter with a threshold of 6% to this scenario would result in filtering of the scenes with surface albedo < 0.1.



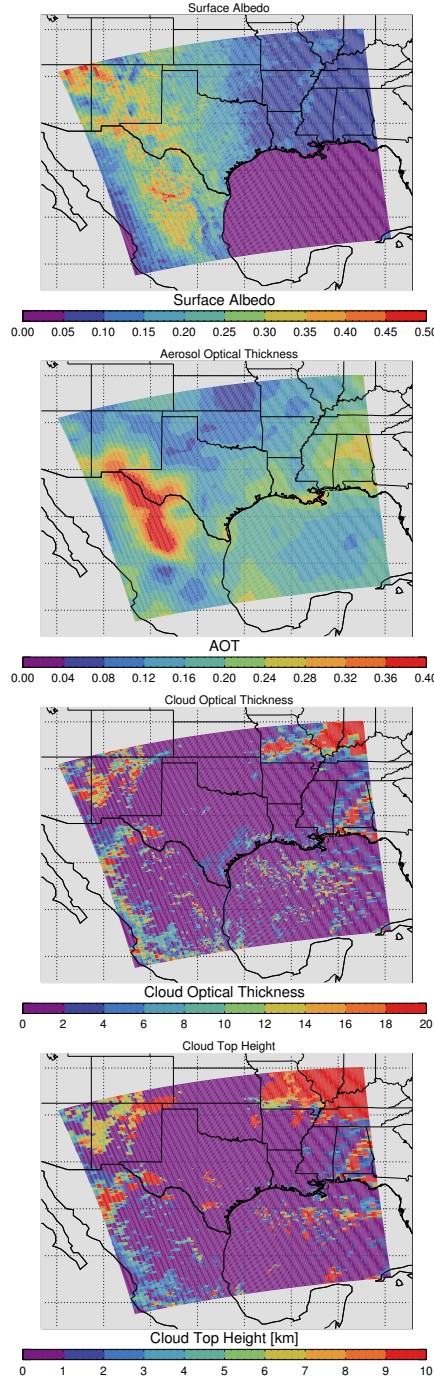

**Fig. 8.** A selection of the input for the realistic scenario simulation. Top: SWIR surface albedo. Second: Aerosol optical thickness at 550 nm. Third: Cloud optical thickness. Bottom: Cloud top height.





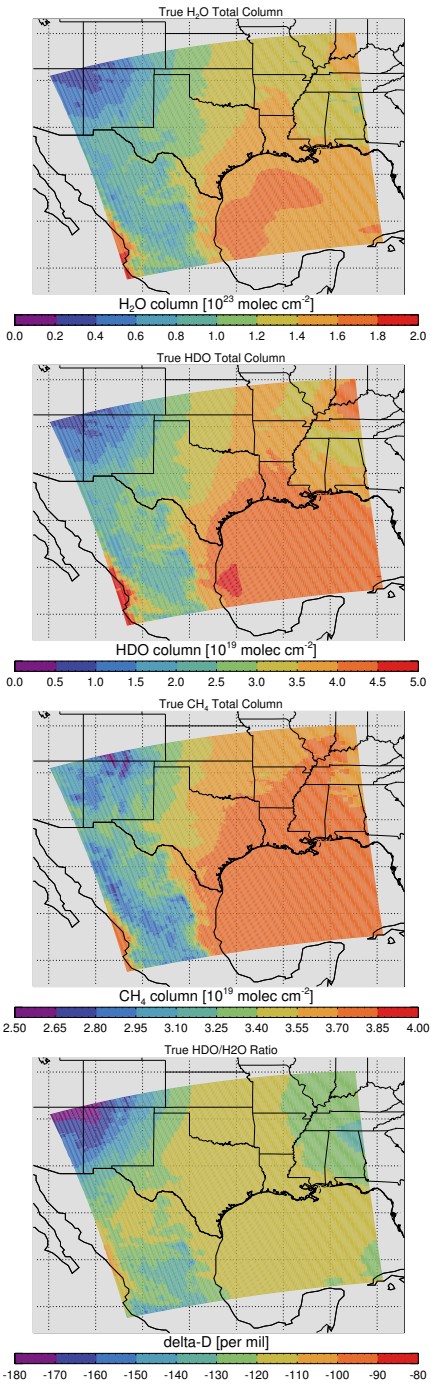

**Fig. 9.** Input total columns for the most important absorbing species for the realistic scenario simulation. Top: $H_2O$. Second: HDO. Third: $CH_4$. Bottom: the resulting total column $HDO/H_2O$ ratio expressed in $\delta D$.





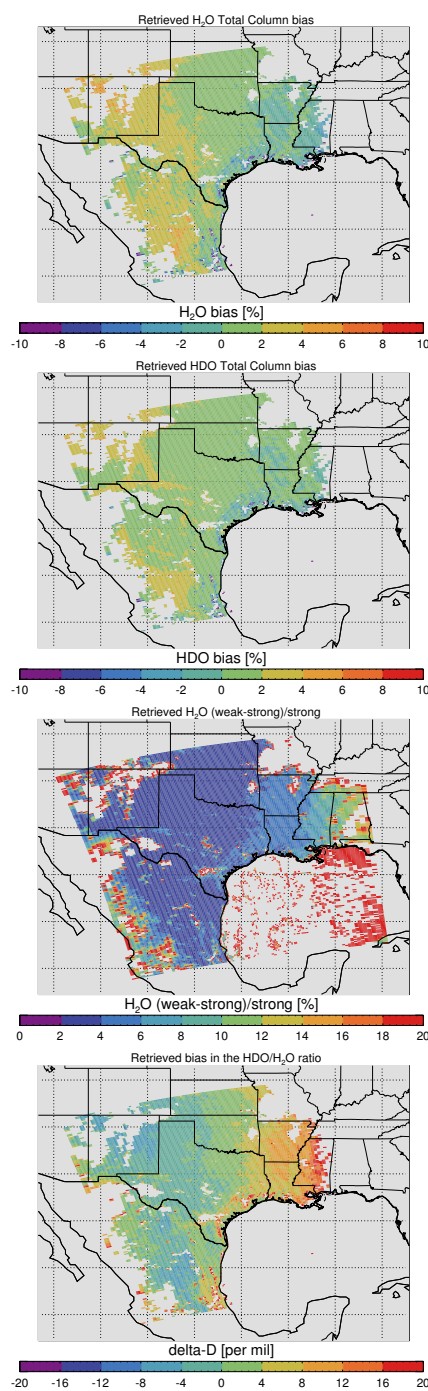

**Fig. 10.** Retrieval biases for water in the realistic scenario simulation. Except for the bottom left panel, the results are cloud filtered using a weak-vs-strong water band threshold of 8%. Top: $H_2O$ bias. Second: HDO bias. Third: relative difference in the weak-vs-strong water bands used for cloud filtering. Bottom: bias in the derived HDO/$H_2O$ ratio.



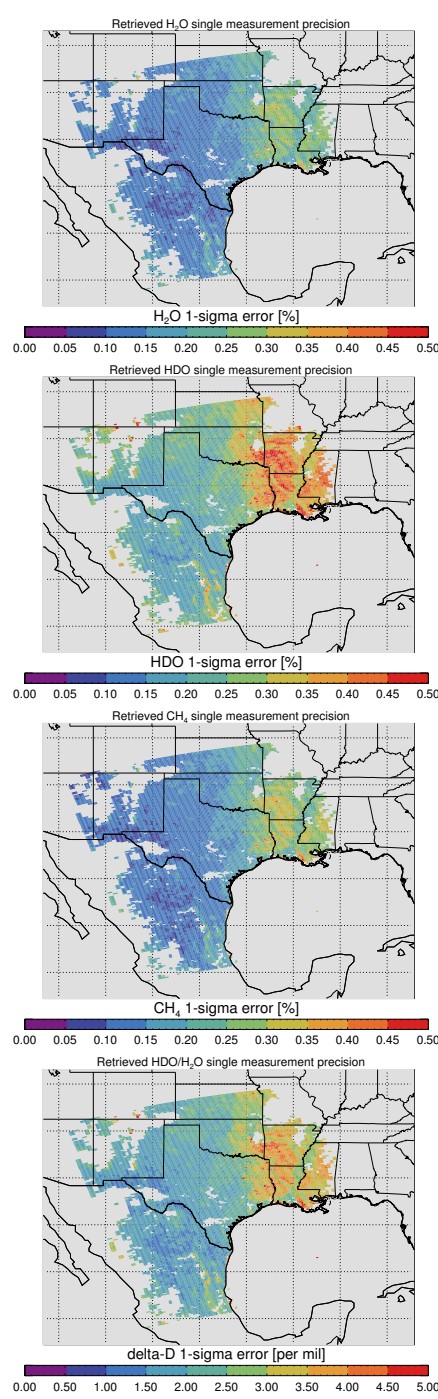

**Fig. 11.** Retrieved, cloud-filtered, single measurement precision of $H_2O$ (top), HDO (second), $CH_4$ (third) and of the derived $HDO/H_2O$ ratio (bottom).