# Peer review of "HDO and $H_2O$ total column retrievals from TROPOMI shortwave infrared measurements"

_Atmospheric Measurement Techniques, 2016_

## Referee Comment (RC1) · Anonymous Referee #1 · 26 May 2016

This is a very well written paper. The paper describes the basic radiative transfer and retrieval theory for estimating HDO and H2O using TROPOMI data, its error analysis, and a discussion of likely interferences that would affect data quality or amplify the errors.

I have read through the paper three times and really have nothing to comment upon with respect to the primary subject.

I would suggest adding some discussion about how different sensitivities between HDO and H2O affect the interpretation of the data (i.e. the Schneider et al. papers and more specifically Schneider et al. AMT 2012) but I leave that up to the discretion of the authors.

[Figure]

**[AMTD](…)**

Interactive
comment

You could also add some discussion on the potential of aircraft validation of these data as there are still unresolved discrepancies between the total column and thermal IR data that could be addressed with aircraft validation. Again, I leave that up to the discretion of the authors, especially since this is an error characterization, not validation, paper.
* * *

---

## Referee Comment (RC2) · Anonymous Referee #2 · 27 May 2016

This paper showcased potential bias and uncertainty of TROPOMI instrument to measure delta-D of total column atmospheric vapor and the overall performance of the retrieval algorithm with a simulation over a North American domain. The resulted performance was acceptable range to study the hydrological cycle in the atmosphere in addition to previous satellite-based instruments. The TROPOMI is planned to launch in the end of 2016.

I found this paper very interesting and well written. Since delta-D in the atmospheric vapor is indeed a useful quantity to understand the atmospheric hydrology and potentially to constrain the atmospheric dynamics through data assimilation, it is very important to improve the instrument. According to the results, the bias and uncertainty of the

retrieved delta-D is only up to 20 permil and 25 permil, respectively. These will lead significant improvement in time and special resolution of the final product due to absence of large number of averaging. So it will be far better than previous instruments, like SCIAMACHY. It is indeed promising.

Only minor revision is needed. Please consider followings:

1. L34: Why don't you add instruments like TES and IASI? For users' point of view, they are all retrieving delta-D to understand the atmospheric hydrology.

2. L40: Not only understanding the hydrological processes, but also constraining the atmospheric circulation is important usefulness of vapor dD observation. Refer Yoshimura et al., 2014, JGR-A.

3. L92: What is SICOR?

4. L105: What is ISRF?

5. L196: What is interference kernel? (with regards to averaging kernel)

6. L250: These parts I was confused. I understood that this cloud filtering was optimized for methane retrieval. Is there any justification that this filtering is also optimized for deltaD retrieval? Why don't you similar figures like Fig 3 for delta-D?

7. L295: Why don't you refer delta18O's performance if you added the delta18O profile? Otherwise there is no reason to add this information in the paper.

8. L374: I don't understand "typical temporal and spatial gradients". Did you mean the range of seasonality or meridional variation? The reasons of those ranges are quite well known. What TROPOMI will add is something like daily variability and/or local (∼100km interval) variations of vapor dD associated synoptic scale weather patterns. If so, the variation interval of at least 10 permil would be needed.

9. L408: What is FWHM?

10. L414: It is likely not true to state that "uncertainties in the input profiles are expect to be random in nature". Particularly for dD, we still don't know the true vertical profile. So there are highly likely to be biased with the current assumption.

11. L423: What is HITRAN? Also, previously S-LINTRAN was used. Why different radiative transfer model is used for this purpose?

12. L451: Probably modeled profiles (especially ECMWF's temperature and humidity and LMDZiso's isotopic profiles) are simpler than the reality. What if the real profiles are complicated (when there are multiple inversions for temperature, vapor, dD)? My guess is that if the profiles are as simple as a-priori profiles, the retrieved values would become closer to the "truth".

---

## Author Comment (AC1) · 29 Jun 2016

***Author comments on:***

**"HDO and H2O total column retrievals from TROPOMI shortwave infrared measurements"**
***By***
**R. A. Scheepmaker, J. aan de Brugh, H. Hu, T. Borsdorff, C. Frankenberg, C. Risi,**
**O. Hasekamp, I. Aben and J. Landgraf**

*Referee comments are in blue italics.*
Author's responses are in black plain text.
Changes made to the manuscript are described in red plain text.

**Referee comments by Anonymous Referee #1**

*This is a very well written paper. The paper describes the basic radiative transfer and retrieval theory for estimating HDO and H2O using TROPOMI data, its error analysis, and a discussion of likely interferences that would affect data quality or amplify the errors. I have read through the paper three times and really have nothing to comment upon with respect to the primary subject.*

We thank the referee for these kind words and for taking the time to thoroughly read our work.

*I would suggest adding some discussion about how different sensitivities between HDO and H2O affect the interpretation of the data (i.e. the Schneider et al. papers and more specifically Schneider et al. AMT 2012) but I leave that up to the discretion of the authors.*

Our retrieval algorithm, being adapted from the SICOR CO retrieval algorithm, follows a different formalism compared to Schneider et al. 2012, making it unfeasible and beyond the scope of our work to apply the techniques from those authors. Instead, we provide the averaging kernels for HDO and H2O for every measurement and encourage future users to take those into account if interpreting the data alongside model data, to account for differences in the assumed profiles. If the data are used as a standalone product (not for model comparison), the sensitivities low in the atmosphere are similar enough not to be a major concern for the overall data use. We therefore choose not to discuss this further in this work.

*You could also add some discussion on the potential of aircraft validation of these data as there are still unresolved discrepancies between the total column and thermal IR data that could be addressed with aircraft validation. Again, I leave that up to the discretion of the authors, especially since this is an error characterization, not validation, paper.*

We are not sure which discrepancies the referee is exactly referring to. Aircraft validation, however, will bring along its own intricacies, for example related to the differences between in-situ sampling and total column measurements. Also, the current aircraft validation campaigns planned for TROPOMI will not be including water isotopologues. We therefore suggest focusing the validation first on total column measurements from ground-based spectrometers. However, we agree with the referee that aircraft validation may be a useful addition to any future validation effort, and therefore have added the following sentence to the last paragraph of the paper:

"Thermal infrared products, such as dD from TES and IASI, also provide useful complementary information due to their different sensitivity. Therefore, aircraft validation may also be valuable, as in-situ measurements could be useful to address any differences between total column and thermal infrared products."

**Referee comments by Anonymous Referee #2**

*This paper showcased potential bias and uncertainty of TROPOMI instrument to measure delta-D of total column atmospheric vapor and the overall performance of the retrieval algorithm with a simulation over*

*a North American domain. The resulted performance was acceptable range to study the hydrological cycle in the atmosphere in addition to previous satellite-based instruments. The TROPOMI is planned to launch in the end of 2016. I found this paper very interesting and well written. Since delta-D in the atmospheric vapor is indeed a useful quantity to understand the atmospheric hydrology and potentially to constrain the atmospheric dynamics through data assimilation, it is very important to improve the instrument. According to the results, the bias and uncertainty of retrieved delta-D is only up to 20 permil and 25 permil, respectively. These will lead significant improvement in time and special resolution of the final product due to absence of large number of averaging. So it will be far better than previous instruments, like SCIAMACHY. It is indeed promising. Only minor revision is needed.*

We are glad that the referee finds our work interesting and promising and thank him/her for the review.

*Please consider followings: 1. L34: Why don't you add instruments like TES and IASI? For users' point of view, they are all retrieving delta-D to understand the atmospheric hydrology.*

The measurement sensitivities of TES and IASI are significantly different from SCIAMACHY and GOSAT such that their time series cannot simply be extended with each other. TROPOMI's sensitivities are more similar to those of SCIAMACHY and GOSAT such that, with the necessary bias corrections from validation studies, trend studies using data from these instruments will be more meaningful. However, we agree with the referee that TES and IASI are equally useful as SCIAMACHY and GOSAT to better understand atmospheric hydrology. Therefore, we have added the following to the outlook (last paragraph of the paper) to revisit the usefulness of thermal infrared data (same addition as mentioned above in reply to Referee #1):

"Thermal infrared products, such as dD from TES and IASI, also provide useful complementary information due to their different sensitivity. Therefore, aircraft validation may also be valuable, as in-situ measurements could be useful to address any differences between total column and thermal infrared products."

*2. L40: Not only understanding the hydrological processes, but also constraining the atmospheric circulation is important usefulness of vapor dD observation. Refer Yoshimura et al., 2014, JGR-A.*

We agree and have changed that sentence accordingly: "A correct understanding of the many interacting processes that control atmospheric humidity, as well as constraining atmospheric circulation, is crucial for General Circulation Models (GCMs) to come to accurate climate projections (Jouzel et al., 1987; Yoshimura et al., 2011; Risi et al., 2012a,b; Yoshimura et al., 2014)."

*3. L92: What is SICOR?*

Replaced with: "Shortwave Infrared CO Retrieval (SICOR) algorithm".

*4. L105: What is ISRF?*

ISRF stands for "Instrument Spectral Response Function" as is mentioned in the same sentence.

*5. L196: What is interference kernel? (with regards to averaging kernel)*

This was described above in lines 180-184: "For $k \neq k'$, $\mathbf{A}_{k,k'}$ describes the interference of the retrieved column $c_k$ with the real trace gas vertical distribution of another trace gas $k'$. For $k=k'$, it is the standard column averaging kernel and we use the more simple notation $\mathbf{A}_k = \mathbf{A}_{k,k}$."
To be more specific we have changed lines 195-196 to: "showing that the contribution of the interference kernel $\mathbf{A}_{k,k'}$ can be interpreted as an error term for every level of the averaging kernel $\mathbf{A}_k$."

*6. L250: These parts I was confused. I understood that this cloud filtering was optimized for methane retrieval. Is there any justification that this filtering is also optimized for deltaD retrieval? Why don't you similar figures like Fig 3 for delta-D?*

The cloud filter is not optimized for methane retrievals, but instead relies on the relative difference in the retrieved column between a weak and strong absorption band. Any absorbing species will do, as long as there is a clear separation between a weak and a strong absorption band. This is why cloud filtering works using either methane or water ($H_2O$), as both have weak and strong absorbing windows (as shown in Fig. 1). HDO is an overall weak absorber compared to $H_2O$ and methane, and therefore not suitable to be used as a cloud filter. It is true that delta-D is very sensitive to clouds (even more so than methane because of the ratio involved), so we simply need the filter that is most sensitive to even the smallest cloud contamination. This is achieved by using either the methane or water bands.

*7. L295: Why don't you refer delta18O's performance if you added the delta18O pro- file? Otherwise there is no reason to add this information in the paper.*

As we have mentioned in lines 136-139, we do account for $H_2^{18}O$ absorption lines, as this improves the fit residuals of the other species, but the absorption lines are not strong enough to result in a sensible retrieval product with reasonable accuracy. Since $H_2^{18}O$ is one of the retrieved species, we think it is appropriate to describe its input profile and show to the reader that we have done everything possible to generate very realistic measurement simulations.

*8. L374: I don't understand "typical temporal and spatial gradients". Did you mean the range of seasonality or meridional variation? The reasons of those ranges are quite well known. What TROPOMI will add is something like daily variability and/or local (~100km interval) variations of vapor dD associated synoptic scale weather patterns. If so, the variation interval of at least 10 permil would be needed.*

With "temporal and spatial gradients" we indeed mean the "range of seasonality and meridional variation". While there are no official requirements for TROPOMI delta-D, we use the range 50-100 permil as an upper limit: the minimum requirement at which the measurements will be useful. We agree with the referee that on smaller scales (both temporal and spatial) a higher accuracy is needed. As we have shown, TROPOMI is able to deliver this higher accuracy, as long as the conditions are cloud free and only moderately affected by aerosol. We thank the referee for pointing this out, and we have changed the last sentence of Section 3.5 accordingly:

"This brings the measurements within the minimum requirement to study, e.g., the range of seasonality and the meridional variation, which are of the order of 50—100 permil. On smaller temporal and spatial scales, such as local daily variability, a higher accuracy is needed, which TROPOMI is able to deliver as long as the conditions are cloud free and only moderately affected by aerosol."

*9. L408: What is FWHM?*

This is now explained as: "full width at half maximum (FWHM)".

*10. L414: It is likely not true to state that "uncertainties in the input profiles are expect to be random in nature". Particularly for dD, we still don't know the true vertical profile. So there are highly likely to be biased with the current assumption.*

Our statement on the quasi-random nature on the input profiles refers to the temperatures and pressures coming from ECMWF. Due to the way ECMWF assimilates observations into its system, we agree that on short spatial and temporal scales (neighbouring ground pixels and six hour time intervals) there will be some correlation between the uncertainties. The further the measurements are apart in terms of time and space, the more this correlation diminishes. Hence the term "quasi-random". Since there is no reason to believe that the ECMWF values on larger scales are systematically biased in a specific direction, we don't expect a systematic bias due to ECMWF uncertainties in the averaged dD either.

For the a priori profiles of the absorbers the situation is different, as we provide averaging kernels. The referee is correct in the sense that dD will be biased if the assumed vertical profiles are different from the true profiles. But the averaging kernels can be used to correct the measured columns to some modelled columns that assumed a different prior profile, or to correct the measured columns to the true vertical profiles, once these are known.

*11. L423: What is HITRAN? Also, previously S-LINTRAN was used. Why different radiative transfer model is used for this purpose?*

HITRAN is an acronym for HIgh-resolution TRANsmission database. We now mention this in the paper as: "For the simulated spectra the parameters from the high-resolution transmission database were used (HITRAN, Rothman et al., 2009)". HITRAN is only used for the spectroscopic parameters and is *not* a radiative transfer model such as S-LINTRAN.

*12. L451: Probably modeled profiles (especially ECMWF's temperature and humidity and LMDZiso's isotopic profiles) are simpler than the reality. What if the real profiles are complicated (when there are multiple inversions for temperature, vapor, dD)? My guess is that if the profiles are as simple as a-priori profiles, the retrieved values would become closer to the "truth".*

This question is basically the topic of Section 4 in the paper ("Sensitivity to prior assumptions"), where we have shown that differences between the real and assumed profiles will lead to various degrees of systematic error in dD. Of course it is unfeasible to test the sensitivity to every possible complicated true profile, so we restricted our analysis to those differences that are expected to occur most frequently (e.g., temperature and pressure differences of the order of ±1K and ±1%, respectively). We do not discuss the impact of differences in the profiles of HDO and H2O, as these differences are characterized by the averaging kernels.

The referee is correct, however, that not all differences between reality and our prior assumptions can be corrected for, which means that the retrieval biases shown in Fig. 10 are likely too optimistic: in reality we will also be dealing with the effects of systematic uncertainties (including uncertain spectroscopy) as discussed in Section 4. To be clear about this in the paper, we added the following sentence after line 497 in Section5:

"The other three panels in Fig. 10 show the remaining biases in total column H2O, HDO and dD after cloud and ocean filtering. One has to keep in mind, however, that any additional bias due to uncertainties in the prior assumptions (as discussed in Sect. 4) is not shown in these figures."

**Additional changes to the manuscript, not suggested by the referees**

In addition to the changes described above, we have also improved figures 3 and 7 by highlighting the 6% curve that will be used for cloud filtering in pink. In the text and figure captions this is now described as:

L253-257:
"We find that with a relative difference in methane absorption <6% (indicated with the pink curve) we effectively filter for clouds and cirrus, as well as for low surface albedo scenes affected by aerosol. For example, not affected by the filter are scenes with a cloud top height <1 km or scenes with a low fraction of higher-level clouds (i.e. everything below or left of the pink curve in the left panel of Fig. 3)."

L359:
"If we take the two-band cloud filter into account (the pink curve coming from Fig. 3) to filter the lowest surface albedos affected by aerosol, we…"

Caption of Fig. 3:
"The pink curve shows the 6% threshold that will be used for filtering."

Caption of Fig. 7:

"The pink curve shows the 6% methane cloud filter threshold from Fig. 3. Applying that filter would result in filtering of the scenes left of the pink line."